# Effect of Animal Age, Postmortem Calcium Chloride Marination, and Storage Time on Meat Quality Characteristics of *M. longissimus thoracis et lumborum* of Buffalo Bulls

**DOI:** 10.3390/foods11203193

**Published:** 2022-10-13

**Authors:** Muawuz Ijaz, Muhammad Hayat Jaspal, Muhammad Usman Akram, Iftikhar Hussain Badar, Muhammad Kashif Yar, Raheel Suleman, Adeel Manzoor, Muhammad Farooq, Sher Ali, Zubair Hussain, Mubarik Mahmood, Abdur Rahman, Rao Sharafat Ali

**Affiliations:** 1Department of Meat Science and Technology, University of Veterinary and Animal Sciences, Lahore 54000, Pakistan; 2Department of Animal Sciences, University of Veterinary and Animal Sciences, Jhang Campus, Jhang 35200, Pakistan; 3Department of Microbiology, FV&AS, The Islamia University of Bahawalpur, Bahawalpur 63100, Pakistan; 4Institute of Food Science and Nutrition, Bahauddin Zakariya University, Multan 61000, Pakistan; 5Department of Clinical Sciences, University of Veterinary and Animal Sciences, Jhang Campus, Jhang 35200, Pakistan; 6Department of Agriculture and Food Technology, Karakorum International University, Gilgit 15100, Pakistan

**Keywords:** buffalo meat, spent buffalo, carabeef, calcium chloride, marination, shelf life, meat science, meat quality, sensory evaluation, tenderness

## Abstract

This study investigated the effect of animal age, calcium chloride marination, and storage time on meat quality characteristics of buffalo bulls to suggest a cost-effective method of improving buffalo meat quality. The current study was designed considering the importance of buffalo meat and the usage of meat from spent buffalo animals in local markets of South Asian countries. A total of 36 animals comprised of 18 young and 18 spent buffalo bulls were selected. After slaughtering and 24 h of postmortem chilling, striploins were separated and cut into 16 steaks and equally divided into two groups, i.e., either marinated with calcium chloride or not. Meat quality characteristics were recorded on 0, 2, 4, 6, 8, and 10 days of storage. The results showed that the pH value of young animals was higher than the value of spent animals and pH was increased over the storage time. Color b*, C*, and h* values were higher in spent animals as compared with the young animals; however, values of colors L* and h* were higher and a* was lower in marinated samples than the values of non-marinated samples. Color a* and C* values were increased and h* was decreased with lengthening the storage time. The meat cooking loss was higher in marinated and the water-holding capacity was higher in non-marinated meat samples. Shear force values were lower in young animals and marinated samples than the values of spent animals and non-marinated meat samples, respectively. Sensory characteristic scores of marinated samples were better than the non-marinated samples. In conclusion, calcium chloride marination can be used to improve the quality characteristics of buffalo meat.

## 1. Introduction

The world buffalo population is estimated to be 198.88 million, spreading across 42 countries, of which 96.4% are distributed across Asia and the major concentration is in India, Pakistan, and China [1]. Kandeepan et al. [2] reported that buffalo meat is preferred for further processing because of its higher lean meat and lower fat contents than cattle meat. It is also reported that buffalo meat has been reported to have the lowest concentration of total lipids [1]. However, in South Asia there is a general public perception that buffalo meat is slightly tough and inferior in quality as compared to cattle meat, which may be due to the fact that buffaloes slaughtered for local market are mostly old [3]. Secondly, there are no specific buffalo breeds used for meat production, and buffalo is mainly raised for dairy production. When mature animals are no longer suitable for dairy production, animals are culled from the herd and slaughtered for meat production. [3,4]. However, there are different strategies to improve meat quality characteristics in postmortem muscles [5].

Since consumers can distinguish between varying degrees of tenderness, therefore, it is a limiting factor in the eating quality of meat that influences consumers’ purchase intentions [5,6,7,8,9]. Modern consumers give more importance to meat flavor, but this quality attribute can be easily manipulated by the use of additives during preparation and various cooking styles, whereas meat tenderness is difficult for consumers to alter [10,11,12]. Consumers are willing to pay a premium price for guaranteed tender meat [10,11]. Destefanis et al. [13] analyzed beef consumer tenderness perception and found that tenderness is a limiting factor for a consumer’s product acceptability due to the wide variability in texture. Except these sensory analysis techniques for meat quality control applications, meat color and tenderness changes/improvement, which are the most important meat quality indicators, are determined using instruments, i.e., Minolta colorimeter and texture analyzer, respectively [14,15]. As mentioned earlier, buffalo meat (carabeef) is being obtained from spent animals, which results in a dark color.

Post-rigor marination with calcium chloride has been shown to increase the tenderness without affecting the other palatability characteristics of meat [5,16,17,18,19]. During the conversion of muscle to meat, cross linkages are formed between actin and myosin filaments, contributing to the toughness of the meat. However, during meat storage, the myofibrillar proteins are degraded by different proteolytic enzymes that ultimately result in tender meat [17,18]. These proteolytic enzymes primarily include calpains, cathepsins, proteasomes, and caspases [17]. Keeping in view the refrigeration costs, calcium-activated tenderization (CAT) can be used to accelerate the tenderization process during storage [18,19]. CAT involves the marination of meat with calcium chloride, in which calcium uses the calpain proteinase system to increase the rate of tenderization or proteolysis that ultimately enhances the meat tenderness [16,17,18]. The natural tenderization process uses µ-calpain only, while m-calpain remains unchanged. However, CAT maximizes the potential of both µ-calpains and m-calpains and thus, enhances the tenderness of meat in a shorter period of refrigeration [16].

Many studies have been performed on the effect of calcium chloride on cattle meat. However, the potential of using calcium chloride on buffalo meat is unexplored. Considering the importance of buffalo meat and its huge potential, this study has been designed to analyze the effect of animal age, calcium chloride marination, along with different storage times on the meat quality characteristics of buffalo bulls. The study will also investigate the potential reduction in storage (ageing) time that can be achieved via calcium chloride marination to obtain acceptable shear force values.

## 2. Materials and Methods

### 2.1. Collections of Samples and Treatments

A total of 36 buffalo (*B. bubalis bubalis*) animals, comprising of 18 young buffalos (with 2 to 2.5 years of age; live weight 263 ± 6.1 kg) and 18 spent buffalos (with 8 to 10 years of age; live weight 486 ± 7.6 kg), bulls were selected. Spent animals refers to mature animals that are no longer suitable for work, or reproduction, and culled from the herd and slaughtered for meat production. All the male animals in intact form were procured from a local animal market. Animals were kept off feed for 12 h and then slaughtered at the meat processing facility of Department of Meat Science and Technology, University of Veterinary and Animal Sciences, Lahore, using halal slaughtering method (PS-3733). The carcasses were not electrically stimulated and the average carcass weight of young bulls was 130 ± 2.8 kg and of spent animals was 245 ± 3.5 kg. After slaughtering, carcasses were shifted to the chiller working at temperature of 0–1 °C with an air velocity of 1 m/s.

After 24 h of postmortem chilling, both left and right sided striploins (*M. longissimus thoracis et lumborum*; LTL) from each carcass were separated and cut into 16 steaks of 3 cm thickness. The striploins were harvested by following the guidelines mentioned in the Handbook of Australian Meat [20], briefly, entire *longissimus* muscles were separated intact, and the surrounding muscles, fat, and connective tissue were removed prior to marination. The sixteen steaks were randomly but equally divided into two (marinated and non-marinated) groups. The marination was performed with food-grade calcium chloride solution (200 mM 5 percent *w*/*w*). The solution was marinated into steaks by using vacuum tumbler (Foodlogistik Fleischereimaschinen GmbH, VV-T-10, Neubrandenburg, Germany) at 0.65 bar for 20 min and allowed to equilibrate for 40 min. Afterwards, marinated and non-marinated steaks were vacuum packed in polyethylene bags (150 × 200, PA/PE 90) using vacuum packing machine (Multivac^®^ Baseline P100, Sepp Hagenuller GmbH & Co.KG, Wolfertschwenden, Germany) and stored for 10 days at 0–1 °C. From each group, six steaks were used to measure the meat quality characteristics on 0, 2, 4, 6, 8, and 10 days of storage and two steaks were used to record the sensory analysis on 2 and 10 days of storage.

### 2.2. pH Measurement

The pH of each steak was measured directly by inserting the electrode of pH meter (WTW, pH 3210 SET 2, Munich, Germany). The pH meter was calibrated with standardized buffers of pH 4 and pH 7. The pH was recorded at three different points along the steaks and the average was obtained and considered as final value on 0, 2, 4, 6, 8, and 10 days of storage.

### 2.3. Color Analysis

Instrumental color of the steaks was measured as described in previous literature [21]. After 1 h of blooming at 0 °C, color was recorded at three different locations of the steaks using chroma meter (Konica Minolta^®^ CR-410, Tokyo, Japan) with C illuminant, 2° standard observer, and 50 mm aperture by avoiding the flakes of fat and connective tissue. The steaks were placed on the white trays in such a manner that muscle fibers on the exposed surface showed perpendicular orientation. The CIE L*, a*, and b* values were recorded to characterize the surface color. Color chroma and hue angle values were calculated as (a*^2^ + b*^2^)^1/2^ and tan^−1^ (b*/a*), respectively. The chroma meter was calibrated each time before use with a standard white tile as instructed by the manufacturer. The total color change (ΔE) was calculated as described previously [22] using following formula:(1)$$\Delta E=\sqrt{\left(L^{*}-L_{0}^{*}\right)+\left(a^{*}-a_{0}^{*}\right)+\left(b^{*}-b_{0}^{*}\right)}$$
where, L^*^_0_, a^*^_0_, and b^*^_0_ are meat color lightness, redness, and yellowness values of young and spent buffalo bulls at day 0.

### 2.4. Cooking Loss and Shear Force Determination

Steaks for cooking loss and shear force analysis were vacuum packed and stored at −20 °C on 0, 2, 4, 6, 8, and 10 days of storage until subsequent analysis. Samples were randomly allocated to three cooking batches and balanced for animal age, marination treatments, and storage days [23]. For cooking loss measurements, steaks were weighed using a digital compact weighing balance (SF-400, 7000 g × 1 g, Shenzhen Electronics Group, Shenzhen, China) and vacuum packed. Then, steaks were placed in a water bath (Memmert WNB45, Munich, Germany) and cooked until they attained the core temperature of 80 °C. Core temperature of steaks was recorded by digital food thermometer (TP101; temperature range of −50 °C to 300 °C). After cooking, steaks were taken out of the water bath and then patted dry with a hand towel and weighed again. The cooking loss was calculated as the difference between the weight of the steaks before and after cooking and the weight of the steaks before cooking and presented in percentages.

For determination of shear force values, cooked steaks were stored overnight at 4 °C. On the following day, strips of 1 × 1 × 4 cm were obtained from each steak parallel to the direction of muscle fibers using scalpel-handle blades. The strips were sheared under the V-slot blade of a texture analyzer (TA.XT plus^®^ texture analyzer, Berkshire, UK) to obtain shear force values that indicated the meat tenderness. The shear force values were recorded in newtons (N/cm^2^) and peak force values from the six strips were recorded and their average was used to calculate the shear force of each sample.

### 2.5. Water-Holding Capacity

Water-holding capacity of both marinated and non-marinated steaks was estimated by compression method in terms of expressible water as described by Grau and Hamm. [24], with little modifications. A 5 g meat sample was placed between filter papers and pressed with a force of 343 N for five minutes using compression machine (YYW-2, Nanjing Soil Instrument, Nanjing, China). The expressible water, a water binding property, was calculated as the difference between the weight of the samples before and after compression and the weight of the steaks before compression and expressed in percentages.

### 2.6. Sensory Analysis

Meat steaks after 2 and 10 days of storage were cooked on a hot plate without addition of pepper and salt to attain the core temperature of 73 °C. The temperature was measured using digital food thermometer (TP101; temperature range of −50 °C to 300 °C). Right after cooking, each steak was cut into 2 cm cubes and immediately served (60 ± 1 °C) to the panelists. The panelists consisted of assistant professors, lecturers, and postgraduate students of the university who were trained for the three months in a 1 h session twice a week (total 24 h). Afterwards, triangle test (ISO 4120:2004) was performed to select the 7 panelists who could differentiate the difference in juiciness, tenderness levels, and off-flavors in cooked meat samples. Samples were tagged anonymously to avoid bias. Panelists were asked to consume the meat pieces and record a score for the samples on 8 point hedonic scale [25]. For odor (8 = extremely beef-like, 7 = very beef-like, 6 = moderately beef-like, 5 = slightly beef-like, 4 = slightly non-beef-like, 3 = moderately non-beef-like, 2 = very non-beef-like, and 1 = extremely non-beef-like), flavor (8 = extremely strong, 7 = very strong, 6 = moderately strong, 5 = slightly strong, 4 = slightly weak, 3 = moderately weak, 2 = very weak, and 1 = extremely weak), texture (8 = extremely tender, 7 = very tender, 6 = moderately tender, 5 = slightly tender, 4 = slightly tough, 3 = moderately tough, 2 = very tough, and 1 = extremely tough), and juiciness (8 = extremely juicy, 7 = very juicy, 6 = moderately juicy, 5 = slightly juicy, 4 = slightly dry, 3 = moderately dry, 2 = very dry, and 1 = extremely dry). Panelists had the facility to rinse their mouth between the samples to avoid carry over taste effect. A total of 10 sensory sessions (including of an initial blank steaks session to familiarize the participants with the scoring system) were conducted with the same participants and each participant evaluated each sample from all treatments.

### 2.7. Statistical Analysis

The statistical analysis was performed using SAS software (version 9.1.3). The data regarding the effect of animal age groups, calcium chloride marination, and storage time on meat pH, color, shear force, cooking loss, and water-holding capacity were analyzed by factorial analysis of variance (ANOVA) using PROC GLM (general linear model). Overall, animal age, marination treatments, and storage times were taken as fixed effects while the carcass and its steaks as random effects. Additionally, cooking batch and shear force test dates were included in random effects for evaluation of cooking loss and shear force. All two-way and three-way interactions were tested and if any of the parameters showed significant effect then all the parameters of those interactions were presented in the manuscript, otherwise presented in Appendix A. The significance level was set at 5% and recorded using Duncan multiple range test.

The following mathematical model was used:Y_ijk_ = μ + α_i_ + β_j_ + γ_k_ + (αβγ)_ijk_ + ε_ijk_(2)
where, Y_ijk_ = observation of dependent variable recorded on ith, jth, and kth treatments; μ = overall population mean; α_i_ = fixed effect of animal age (i = 1, 2); β_j_ = fixed effect of marination treatment (j = 1,2); γ_k_ = fixed effect of storage time (k = 1, 2, 3, 4, 5, 6); (αβγ)_ijk_ = interactions effects of animal age, marination treatment, and storage time; and ε_ijk_ = residual effect.

Data for sensory analysis of odor, flavor, texture, and juiciness were corrected by linear mixed model (PROC GIMMIX) and included panelist and sessions as fixed and random terms in the model, respectively [22]. Variables were normally distributed using histogram and QQ part of the residual. The panelist and session effects were excluded from the final model as the difference between them was non-significant. The Tukey–Kramer test was used to compare the mean values and *p*-value was set at 5%. The data were presented as the mean ± standard error.

## 3. Results and Discussion

### 3.1. pH

The effect of animal age, calcium chloride marination, and storage time on the meat pH of buffalo bulls is shown in Table 1. The results showed that the pH of young animals was higher as compared with the spent animals (*p* < 0.05). The pH was slightly increased by calcium chloride marination; however, the difference was non-significant (*p* > 0.05). Among storage time, the pH was increased over the storage time and the lowest value was recorded at day 0 and the highest value was noted at day 10 of storage (*p* < 0.05). Two-way interactions of age groups and storage days were significantly different and shown in Table 2 (*p* < 0.05). In this interaction, the highest pH value was recorded at day 10 in young animals and the lowest values were noted at days 0, 2, and 4 in spent animals as compared with the all-other treatments.

Lower pH values of the spent animals could be due to their increased ante-mortem glycogen contents, which favors the post-mortem glycolysis and accumulation of lactic acid than that of the young animals [7]. A non-significant increase in pH due to calcium chloride marination was also observed by Perez et al. [19]. Though contradictory results were obtained by Dikeman et al. [26], who infused a solution of calcium chloride in pre-rigor beef carcasses and observed a rapid decline in pH in these carcasses (*p* < 0.05). The pH increased till day 10, which is similar to the pattern of pH observed by Karabagias et al. [27]. This increase in pH during storage is attributed to protein breakdown, which results in the production of free amino acids leading to the formation of ammonia and amines, compounds of alkaline nature. A slight decrease in pH was observed by Stanisic et al. [28] during storage until the 7th day, but this was not statistically significant. The highest pH value at day 10 in young animals could be due to more protein breakdown, and the formation of an alkaline nature in these animals [27].

### 3.2. Color

The effect of animal age, calcium chloride marination, and storage time on the meat color of buffalo bulls is shown in Table 1. The color lightness (L*) value of young animals was similar with the value of spent animals. However, the L* value of calcium chloride-marinated meat was found to be significantly higher compared to non-marinated meat (*p* < 0.05). Color L* values remained non-significant over the storage time. The color redness (a*) value was not affected by the animal age and the a* value of young animals was similar with the spent animals. Calcium chloride-marinated samples showed significantly lower redness value than that of the non-marinated samples (*p* < 0.05). The storage period had a significant effect on the a* value and samples at day 10 showed significantly lower a* values as compared with the values at day 0.

Meat samples from spent buffalo bulls showed significantly higher yellowness (b*) values than samples from young buffalo bulls (*p* < 0.05). Marination did not affect the b* value of the samples and values of both age groups were similar with each other. Similarly, b* values remined unchanged throughout the storage period (*p* > 0.05). The meat chroma (C*) value was significantly higher in spent animals than the value of young animals (*p* < 0.05). The C* value was not affected by calcium chloride marination and similar values were recorded between marinated and non-marinated meat samples. During storage, C* values were decreased with lengthening the storage time and the C* value at day 0 was higher than the value at day 10.

The hue (h*) value was higher in spent animals as compared with the values of young animals. The h* value of marinated meat samples was higher than that of the non-marinated samples. The storage period had a significant effect on the h* value and h* decreased from day 0 to day 10 (*p* < 0.05). The change in color (ΔE) value was not affected by the animal age groups. However, the ΔE value of the marinated meat samples was higher than the value of the non-marinated meat samples. The ΔE was decreased with the increase in storage time and the ΔE value at day 10 was higher than the value at day 0. The two-way interaction effects of animal age groups and storage days of meat a*, b*, C*, and ΔE were significantly different and shown in Table 2 (*p* < 0.05). The highest a* value was recorded at day 0 in spent animals and the lowest values were noted at days 6, 8, and 10 in spent animals and at day 0 and 2 in the young animals. Similarly, the highest b* value was recorded at day 0 in spent animals and the lowest values were noted at days 0 and 2 in the young animals. The C* value was higher at day 0 in spent animals and ΔE was higher at day 10 in spent animals as compared with the all-other treatments (*p* < 0.05).

The increase in the L* value due to calcium chloride marination could be attributed to the structural changes in the muscle proteins during marination of the meat. These structural changes could increase the light scattering that ultimately increases the L* value. Secondly, the increase in lightness values may be due to less mitochondrial activity in muscle cells of marinated meat samples, leading to a lower oxygen consumption rate (OCR) and an increased oxygen penetration depth (OPD). This results in more oxygenation of myoglobin causing the development of a thick oxy-myoglobin layer that increases the color lightness [29]. Calcium chloride marination caused a reduction in the a* value, which is in agreement with the results of Jaturasitha et al. [30]. The reduction in redness could be attributed to accelerated proteolysis as a result of calcium chloride marination. However, an increase in redness values during storage was observed by Mancini and Ramanathan [29]. The higher a* value at day 0 in spent animals could be due to more myoglobin contents in these older age animals as compared with the young animals [21].

Higher b* (yellowness) values in spent animals could be due to the deposition of a yellow carotenoid pigment in these animals as compared with the young animals [21]. Marination did not affect the color b* and C* values and these results are similar with the findings of Rajagopal et al. [31]. The highest b* value at day 0 in spent animals and the lowest at days 0 and 2 in the young animals could be due to a higher deposition of carotenoid pigment with the increase in animal age [21]. Similar with the results of L*, the h* value was increased due to the marination and the same results were obtained by Perez et al. [19].

Color C* and h* were higher in spent animals. The C* and h* are derived from a* and b* values and these are influenced by the relative proportion of myoglobin and functions of mitochondria. The C* represents the color intensity and increased with the increase in OPD [32]. It is also related with the function of mitochondria and increases with the decrease in OCR. The h* value is positively linked with OPD and negatively related with the OCR [32]. The a* and C* values were decreased from day 0 to day 10 in the present study. Similar results were obtained by Naveena et al. [33]. However, an increase in C* was observed by Stanisic et al. [28] during storage. The higher C* value at day 0 in spent animals could be due to more OPD or reduce the level of OCR in spent animals at the start of storage [32]. The h* values increased with the storage duration, which is in agreement with the results of [28]. The change in color (ΔE) of meat is associated with the change in myoglobin form or formation of metmyoglobin as well as non-enzymatic reactions between the products of oxidation and amines in meat [34]. In the current study, higher ΔE values in marinated samples could be due to the fact that calcium chloride marination may decrease the OPD and increase the OCR, which ultimately led to the increase in ΔE values in marinated samples [34,35]. The increase in ΔE at day 10 in spent animals may be due to a decrease in OPD and an increase in OCR values in spent animals.

### 3.3. Shear Force and Cooking Loss

The effect of animal age, calcium chloride marination, and storage time on the shear force and cooking loss of buffalo bulls is shown in Table 3. Results of the present study showed that the shear force value of spent animals was higher compared with the value of young animals (*p* < 0.05). Furthermore, calcium chloride marination significantly reduced the shear force value and it was lower in marinated samples than the value of non-marinated meat samples (*p* < 0.05). Shear force values decreased continuously over the storage time and the highest value was recorded at day 0 and the lowest value was noted at day 10 of the storage. The results of the cooking loss indicated that there was a non-significant difference between the cooking loss of young and spent buffalo groups. However, samples marinated with calcium chloride showed a higher cooking loss than non-marinated samples (*p* < 0.05). The storage period did not have any effect on the cooking loss. Two-way interactions of cooking loss between age groups and treatments, of shear force between age groups and storage days and of both cooking loss and shear force between treatments and storage days were significantly different and shown in Table 4 (*p* < 0.05). The cooking loss of marinated samples in both young and spent animals were similar but higher than the values of non-marinated samples in the interaction of age groups and treatments. Between age groups x storage days, the highest shear force value was recorded at day 0 in the spent animals and the lowest value was noted at day 10 in the young animals (*p* < 0.05). Similarly, between treatment x storage days, the highest shear force value was recorded at day 0 in the non-marinated samples and the lowest value was noted at day 10 in the marinated meat samples (*p* < 0.05). Between three-way interactions, the highest shear force value was recorded at day 0 in spent animals-non-marinated meat samples and the lowest value was noted at day 10 in young animals-marinated meat samples.

The increase in shear force value with the increase in animal age could be due to an increase in the amount of connective tissue in the spent animals [21]. These results are in agreement with the reports of Schonfeldt and Strydom [36], who observed the effect of three different age groups on tenderness of South African beef. They attributed the increase in shear force values in older animals to the increase in connective tissue with the age of animal.

The decrease in shear force value due to marination was also reported in previous studies [37,38,39,40,41]. The decrease in shear force values could be credited to the activation of the calpain proteolytic system due to marination with calcium chloride. The natural tenderization process uses µ-calpain only, while m-calpain remains unchanged. However, calcium-activated tenderization maximizes the potential of both calpains (µ-calpain and m-calpain) and enhances the tenderness of meat [31,42]. On the other hand, it was reported that an infusion of lamb carcasses with 0.15 M calcium chloride increases the shear force values to 13% compared to non-infused carcasses [38]. An increase in shear force values after pre-rigor calcium chloride marination could be the result of the calcium-induced shortening of muscle fibers.

The decrease in the shear force value with the increase in storage time may be due to the process of tenderization [21]. Purslow [40] reported similar observations and concluded that the decrease in the shear force value with the increase in storage time is mainly due to the decrease in strength of intramuscular connective tissue during storage. As the storage time increases, the pH also increases, which favors the activity of calpains, resulting in increased tenderness [27,43,44]. Similarly, in interaction effects, the lowest shear force value at day 10 in the young animals-marinated meat samples could be due to less connective tissue in the young animals, the activation of the calpain proteolytic system due to marination, and the action of calpains on the myofibrillar proteins over the storage time [21,27,29].

An increased cooking loss in marinated samples was also observed by Morgan et al. [45]. An increase in the cooking loss in marinated samples is attributed to the loss of calcium chloride solution during cooking. However, Lansdell et al. [37] reported a non-significant difference in the cooking loss of marinated and non-marinated samples. Between the interaction of age groups and treatments, the cooking loss of marinated samples in both young and spent animals were similar but higher than the values of non-marinated samples that could be due to the loss of calcium chloride solution along with water molecules in the marinated meat samples during cooking [45].

### 3.4. Water-Holding Capacity

Expressible water depicts the water-holding capacity of meat. The effect of animal age, calcium chloride marination, and storage time on the expressible water of buffalo bulls is shown in Table 3. Results of the present study showed that there was a non-significant difference between the expressible water or water-holding capacity of young and spent animals. However, calcium chloride marination increased the expressible water or in other words decreased the water-holding capacity and value of expressible water of marinated meat samples was higher than the value of non-marinated meat samples (*p* < 0.05). There was a non-significant difference of water-holding capacity among 0, 4, 6, 8, and 10 days of storage. However, the water-holding capacity at day 2 was lower than the value at all other storage days. All the two-way and three-way interactions of animal age groups, treatments, and storage days of expressible water were non-significant (Table 4).

Calcium chloride marination resulted in a reduced water-holding capacity, as observed by Gerelt et al. [35]. A reduced water-holding capacity is expected as calcium interrupts the water absorption, or it can be due to the addition of extra weight as a fluid. The reduction in water-holding capacity as a result of calcium chloride marination was also observed by Dikeman et al. [26]. However, no significant difference in the water-holding capacity of marinated and control samples was documented by Rajagopal et al. [31]. On the other hand, an increase in the water-holding capacity with the increase in storage was also observed by Kristensen and Purslow [46]. During storage, the cytoskeleton degrades and thus slowly removes the shrinkage of muscle fibers, which stops the flow into the extracellular space and increases the water-holding capacity.

### 3.5. Sensory Evaluation

The effect of animal age, calcium chloride marination, and storage time on sensory attributes (odor, flavor, texture, and juiciness) of buffalo bulls is shown in Table 5. Results of the sensory evaluation showed that age had a significant effect on beef odor and texture and young animals showed lower odor and higher texture scores as compared with the spent animals (*p* < 0.05). Marination significantly affects all the sensory attributes and marinated meat samples showed higher beef odor, flavor, texture, and juiciness values than the values of non-marinated samples (*p* < 0.05). Furthermore, the flavor score was decreased with the increase in storage time and its value was higher at day 2 than the score at day 10 of storage. However, odor texture and juiciness were not changed over the storage period. Two-way interactions between age groups and treatments (sensory odor, flavor, and texture) and between treatments and days (odor) were significant and three-way interactions of odor were significantly different (*p* < 0.05) and shown in Table 6. Between age groups and treatments interactions, odor and flavor values were higher in spent animals-marinated meat samples (*p* < 0.05). However, the texture score was higher in young animals-marinated meat samples. The odor score was higher in marinated meat samples at day 2 than the score of non-marinated at day 2 between the interactions of treatment and storage days. In three-way interactions, the odor score of spent animals-marinated meat samples at day 2 was highest as compared with the other treatments (*p* < 0.05).

The results obtained in this study showed that calcium chloride marination produces extreme odor and strong flavor. Perez et al. [19] found that calcium chloride marination produces a bitter flavor and a poor after-taste that were perhaps due to the long marination time of 48 h. Results of the present study showed that there was a significant improvement in sensory texture scores of marinated samples that could be due to the activation of the calpain system by the calcium marination, which ultimately degrades the myofibrillar proteins and improves the tenderness [35,36,39]. The increase in juiciness scores in the marinated meat samples could be due to a lower capacity of these samples to hold the water at the time of chewing [35]. The sensory scores obtained in this study after marination are similar with the findings of Diles et al. [47] and Carr et al. [48], they also reported higher scores for texture and juiciness in calcium chloride-marinated samples. Contrary to it, Clare et al. [49] found that the use of calcium chloride marination showed a minimal effect on the sensory properties of meat.

## 4. Conclusions

In conclusion, postmortem marination with calcium chloride could increase the color lightness and decrease the meat shear force values. Furthermore, marination did not have any adverse effects on the sensory properties of meat. Therefore, postmortem marination with calcium chloride could be used to improve the meat quality characteristics of young and spent buffalo bulls without any negative effect on the sensory traits. Consequently, the cost of production can be decreased with the use of calcium chloride marination as compared to the traditional ageing method that requires longer storage durations to improve the meat quality of buffalo bulls, which is difficult to promote in energy- and technology-deficient developing countries. The major limitation of the current study is a lack of focus on the molecular mechanisms; therefore, further studies should explore the mechanistic effect of using calcium chloride to delineate the pathways involved behind the improvement in meat quality attributes.

## Figures and Tables

**Table 1 foods-11-03193-t001:** Effect of age groups, treatments, and storage time on meat pH and color parameters (L*, a*, b*, C*, h*, and ΔE) of *M. longissimus thoracis et lumborum* of buffalo bulls.

	pH	L*	a*	b*	C*	h*	ΔE
Age groups							
Young	5.77 ^a^	46.49	16.41	6.24 ^b^	17.58 ^b^	20.70 ^b^	2.40
Spent	5.60 ^b^	45.92	16.47	7.92 ^a^	18.39 ^a^	25.40 ^a^	2.35
SE *	0.01	0.22	0.23	0.10	0.22	0.35	0.12
Treatments							
Marinated	5.70	46.78 ^a^	16.06 ^b^	7.20	17.76	24.05^a^	2.38 ^a^
Non-Marinated	5.68	45.64 ^b^	16.81 ^a^	6.96	18.21	22.05 ^b^	2.25 ^b^
SE *	0.01	0.22	0.23	0.10	0.22	0.35	0.15
Storage time (days)							
0	5.62 ^c,d^	47.16	18.40 ^a^	7.41	19.95 ^a^	20.93 ^b^	NA ^#^
2	5.60 ^d^	46.36	16.36 ^b^	7.05	18.01 ^b^	22.81 ^a,b^	2.22 ^c^
4	5.67 ^b,c^	46.31	16.62 ^b^	7.10	18.13 ^b^	23.07 ^a,b^	1.99 ^c^
6	5.71 ^b^	46.01	16.39 ^b^	7.04	17.82 ^b^	23.42 ^a,b^	2.35 ^c^
8	5.73 ^a,b^	45.59	16.21 ^b,c^	7.13	17.79 ^b,c^	23.24 ^a,b^	2.71 ^b,c^
10	5.79 ^a^	45.81	14.64 ^c^	6.74	16.20 ^c^	24.80 ^a^	4.05 ^a^
SE *	0.02	0.38	0.40	0.18	0.39	0.62	0.22
ANOVA (*p*-value)							
Age groups	0.000	0.072	0.865	0.000	0.014	0.000	0.061
Treatments	0.086	0.000	0.027	0.105	0.170	0.000	0.012
Days	0.000	0.080	0.000	0.241	0.000	0.002	0.031

Means in the same column with different small letters (a,b,c) are significantly different (*p* < 0.05) within age groups, treatments, or storage time (days). * SE: standard error. ^#^ NA: not applicable.

**Table 2 foods-11-03193-t002:** Interaction effects of age groups, treatments, and storage days on meat pH, color parameters (L*, a*, b*, C*, h*, and ΔE) of *M. longissimus thoracis et lumborum* of buffalo bulls.

		pH	L*	a*	b*	C*	h*	ΔE
Age groups × Days								
Young	Day 0	5.70 ^b,c,d^	47.07	15.08 ^b,c,d^	5.07 ^f^	16.00 ^d,e^	18.29	4.06 ^b,c^
	Day 2	5.70 ^b,c,d^	46.92	15.59 ^b,c,d^	5.57 ^e,f^	16.59 ^c,d,e^	19.55	3.37 ^c,d^
	Day 4	5.80 ^a,b^	46.67	16.82 ^b,c^	6.46 ^d,e^	18.06 ^b,c,d^	20.95	1.91 ^e,f^
	Day 6	5.80 ^a,b^	45.98	17.18 ^b,c^	6.49 ^d,e^	18.29 ^b,c,d^	20.72	1.93 ^e,f^
	Day 8	5.78 ^a,b,c^	46.04	17.59 ^b^	7.20 ^c,d^	19.01 ^b,c^	22.28	1.40 ^f^
	Day 10	5.86 ^a^	46.29	16.22 ^b,c^	6.67 ^c,d,e^	17.53 ^b,c,d^	22.40	2.46 ^e^
Spent	Day 0	5.55 ^e,f^	47.25	21.73 ^a^	9.76 ^a^	23.90 ^a^	23.65	4.08 ^b,c^
	Day 2	5.49 ^f^	45.80	17.14 ^b,c^	8.53 ^a,b^	19.42 ^b^	26.06	2.17 ^e^
	Day 4	5.55 ^e,f^	45.96	16.43 ^b,c^	7.73 ^b,c^	18.19 ^b,c,d^	25.19	2.33 ^e^
	Day 6	5.62 ^d,e^	46.05	15.60 ^b,c,d^	7.59 ^b,c,d^	17.35 ^b,c,d,e^	26.12	3.02 ^d,e^
	Day 8	5.69 ^c,d^	45.15	14.83 ^c,d^	7.06 ^c,d^	16.57 ^c,d,e^	24.20	4.11 ^b,c^
	Day 10	5.72 ^b,c,d^	45.33	13.07 ^d^	6.81 ^c,d^	14.87 ^e^	27.19	5.66 ^a^
	SE *	0.02	0.54	0.57	0.26	0.55	0.87	0.16
ANOVA (*p*-value)								
Age groups × Treatments		0.188	0.747	0.106	0.902	0.152	0.090	0.325
Age groups × Days		0.011	0.765	0.000	0.000	0.000	0.174	0.021
Treatment × Days		0.968	0.261	0.458	0.224	0.307	0.137	0.264
Age groups × Treatment × Days		0.464	0.239	0.184	0.238	0.146	0.806	0.159

Means in the same column with different small letters (a, b, c, d, e, and f) are significantly different (*p* < 0.05). * SE: standard error.

**Table 3 foods-11-03193-t003:** Effect of age groups, treatments, and storage time on shear force (SF), cooking loss (CL%), and expressible water (EW%) of *M. longissimus thoracis et lumborum* of buffalo bulls.

	SF (N)	CL%	EW% ^#^
Age groups			
Young	30.83 ^b^	40.30	13.58
Spent	37.84 ^a^	40.41	13.31
SE *	0.10	0.24	0.20
Treatments			
Marinated	32.87 ^b^	41.87 ^a^	16.26 ^a^
Non-Marinated	35.80 ^a^	38.84 ^b^	10.63 ^b^
SE *	0.10	0.24	0.20
Storage time (days)			
0	39.44 ^a^	40.18	13.13 ^b^
2	37.27 ^b^	40.86	15.25 ^a^
4	34.91 ^c^	39.95	13.70 ^b^
6	33.11 ^d^	40.78	12.93 ^b^
8	31.33 ^e^	40.25	13.26 ^b^
10	29.89 ^f^	40.09	12.39 ^b^
SE *	0.18	0.43	0.34
ANOVA (*p*-value)			
Age groups	0.000	0.764	0.341
Treatments	0.000	0.000	0.000
Days	0.000	0.580	0.000

Means in the same column with different small letters (a, b, c, d, e, and f) are significantly different (*p* < 0.05) within age groups, treatments, or storage time (days). * SE: standard error. ^#^ EW depicts the water-holding capacity of meat.

**Table 4 foods-11-03193-t004:** Interaction effects of age groups, treatments, and storage days on shear force (SF), cooking loss (CL %), and expressible water (EW %) of *M. longissimus thoracis et lumborum* of buffalo bulls.

		SF (N)	CL%	EW% ^#^
Age groups × Treatments				
Young	Marinated	29.37	41.45 ^a^	16.56
	Non-Marinated	32.29	39.15 ^b^	10.60
Spent	Marinated	36.37	42.29 ^a^	15.96
	Non-Marinated	39.31	38.52 ^b^	10.65
	SE *	0.14	0.35	0.28
Age groups × Days				
Young	Day 0	35.21 ^d,e^	39.25	13.81
	Day 2	33.72 ^f,g^	41.13	15.56
	Day 4	31.61 ^h^	41.24	13.80
	Day 6	29.92 ^i^	40.11	13.06
	Day 8	28.11 ^j^	39.96	13.13
	Day 10	26.44 ^k^	40.13	12.12
Spent	Day 0	43.67 ^a^	41.12	12.46
	Day 2	40.81 ^b^	40.58	14.95
	Day 4	38.22 ^c^	38.67	13.59
	Day 6	36.31 ^d^	41.46	12.80
	Day 8	34.66 ^e,f^	40.55	13.39
	Day 10	33.35 ^g^	40.06	12.66
	SE *	0.28	0. 66	0. 58
Treatment × Days				
Marinated	Day 0	38.68 ^b^	42.69 ^a^	15.29
	Day 2	36.20 ^c^	42.31 ^a^	18.24
	Day 4	33.40 ^e^	42.25 ^a,b^	16.81
	Day 6	31.45 ^f^	41.48 ^a,b,c^	15.70
	Day 8	29.45 ^g^	41.31 ^a,b,c^	16.22
	Day 10	28.04 ^h^	41.18 ^a,b,c^	15.31
Non-Marinated	Day 0	40.20 ^a^	37.68 ^d^	10.97
	Day 2	39.33 ^b^	39.40 ^b,c,d^	12.27
	Day 4	36.43 ^c^	37.65 ^d^	10.58
	Day 6	34.78 ^d^	40.09 ^a,b,c,d^	10.16
	Day 8	33.31 ^e^	39.19 ^c,d^	10.30
	Day 10	31.74 ^f^	39.01 ^c,d^	9.46
	SE *	0.25	0. 70	0.49
Age groups × Treatment × Days				
Young Marinated	Day 0	34.34 ^h^	42.20	16.88
	Day 2	32.75 ^i,j^	42.25	18.87
	Day 4	30.12 ^l^	42.82	16.86
	Day 6	28.10 ^m^	40.28	15.72
	Day 8	26.17 ^n^	40.23	16.08
	Day 10	24.80 ^o^	40.94	14.97
Young Non-Marinated	Day 0	36.09 ^f,g^	36.30	10.74
	Day 2	34.71 ^h^	40.01	12.26
	Day 4	33.11 ^i^	39.66	10.75
	Day 6	31.75 ^j,k^	39.95	10.41
	Day 8	30.05 ^l^	39.70	10.18
	Day 10	28.08 ^m^	39.33	9.28
Spent Marinated	Day 0	43.03 ^b^	43.18	13.71
	Day 2	39.67 ^b^	42.38	17.62
	Day 4	36.68 ^f^	41.69	16.76
	Day 6	34.81 ^h^	42.69	15.70
	Day 8	32.74 ^i,j^	42.41	16.36
	Day 10	31.29 ^k^	41.42	15.66
Spent Non-Marinated	Day 0	44.33 ^a^	39.07	11.22
	Day 2	41.96 ^c^	38.80	12.29
	Day 4	39.77 ^b^	35.65	10.43
	Day 6	37.81 ^e^	40.24	9.92
	Day 8	36.58 ^f^	38.69	10.44
	Day 10	35.41 ^g,h^	38.71	9.66
	SE *	0.36	0.86	0.69
ANOVA (*p*-value)				
Age groups × Treatments		0.945	0.038	0.256
Age groups × Days		0.001	0.027	0.469
Treatment × Days		0.000	0.017	0.439
Age groups × Treatment × Days		0.000	0.369	0.257

Means in the same column with different small letters (a–o) are significantly different (*p* < 0.05). * SE: standard error. ^#^ EW depicts the water-holding capacity of meat.

**Table 5 foods-11-03193-t005:** Effect of age groups, treatments, and storage time on sensory attributes ^#^ (odor, flavor, texture, and juiciness) of *M. longissimus thoracis et lumborum* of buffalo bulls.

	**Odor**	**Flavor**	**Texture**	**Juiciness**
Age groups				
Young	5.80 ^b^	5.90	5.95 ^a^	5.79
Spent	6.29 ^a^	6.05	5.62 ^b^	5.63
SE *	0.07	0.06	0.05	0.06
Treatments				
Marinated	6.17 ^a^	6.14 ^a^	6.25 ^a^	5.85 ^a^
Non-Marinated	5.91 ^b^	5.81 ^b^	5.41 ^b^	5.56 ^b^
SE *	0.07	0.06	0.05	0.06
Storage time (days)				
2	6.10	6.07 ^a^	5.79	5.65
10	5.99	5.88 ^b^	5.88	5.76
SE *	0.07	0.06	0.05	0.06
ANOVA (*p*-value)				
Age groups	0.000	0.104	0.037	0.098
Treatments	0.014	0.001	0.000	0.003
Days	0.298	0.050	0.617	0.257

Means in the same column with different small letters (a, b) are significantly different (*p* < 0.05) within age groups, treatments, or storage time (days). * SE: standard error. # 1 = extremely non-beef-like odor, weak flavor, tough texture, and dry in terms of juiciness; 8 = extremely beef-like odor, strong flavor, tender texture, and juicy.

**Table 6 foods-11-03193-t006:** Interaction effects of age groups, treatments, and storage days on sensory attributes ^#^ (odor, flavor, texture, and juiciness) of *M. longissimus thoracis et lumborum* of buffalo bulls.

		Odor	Flavor	Texture	Juiciness
Age groups × Treatments					
Young	Marinated	5.70 ^b^	5.88 ^b^	6.43 ^a^	5.86
	Non-Marinated	5.90 ^b^	5.91 ^b^	5.41 ^c^	5.71
Spent	Marinated	6.65 ^a^	6.40 ^a^	6.08 ^b^	5.85
	Non-Marinated	5.93 ^b^	5.71 ^b^	5.41 ^c^	5.41
	SE	0.10	0.09	0.08	0.09
Treatment × Days					
Marinated	Day 2	6.35 ^a^	6.30	6.18	5.88
	Day 10	6.00 ^a,b^	5.98	6.33	5.83
Non-Marinated	Day 2	5.85 ^b^	5.85	5.38	5.43
	Day 10	5.98 ^a,b^	5.78	5.45	5.70
	SE	0.10	0.09	0.08	0.09
Age groups × Treatment x Days					
Young Marinated	Day 2	5.76 ^b,c^	5.96	6.06	5.86
	Day 10	5.63 ^c^	5.80	6.10	5.86
Young Non-Marinated	Day 2	6.00 ^b,c^	5.90	5.43	5.63
	Day 10	5.80 ^b,c^	5.93	5.40	5.80
Spent Marinated	Day 2	6.93 ^a^	6.63	6.60	5.90
	Day 10	6.36 ^a,b^	6.16	6.26	5.80
Spent Non-Marinated	Day 2	5.70 ^c^	5.80	5.33	5.23
	Day 10	6.16 ^b,c^	5.63	5.50	5.60
	SE	0.14	0.13	0.11	0.13
ANOVA (*p*-value)					
Age groups × Treatments		0.000	0.000	0.037	0.139
Age groups × Days		0.575	0.199	0.617	0.794
Treatment × Days		0.021	0.199	0.194	0.098
Age groups × Treatment x Days		0.009	0.797	0.090	0.433

Means in the same column with different small letters (a, b, and c) are significantly different (*p* < 0.05). * SE: standard error. ^#^ 1 = extremely non-beef-like odor, weak flavor, tough texture, and dry in term of juiciness; 8 = extremely beef-like odor, strong flavor, tender texture, and juicy.

## Data Availability

Not applicable.

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
