# Peer review of "Effect of Animal Age, Postmortem Calcium Chloride Marination, and Storage Time on Meat Quality Characteristics of M. longissimus thoracis et lumborum of Buffalo Bulls"

_foods, 2022, doi:10.3390/foods11203193_

Round 1

Reviewer 1 Report

The authors are advised to expand the Introduction a bit by including more recent references and research findings on the topic, especially on the use of color analysis and shear force evaluation techniques for such food quality control applications.

It is suggested to divide the Table 1 into two tables with separate results for pH values and color parameters.

It would be better to present the results (for eg. color parameters) by bar graphs or some other plots for better visualization.

Following papers are advised to cite in order to enhance the quality of the manuscript:

doi: 10.1007/s13197-012-0882-x

DOI: 10.1109/ISOEN.2017.7968861

doi: 10.3390/ani11072111

What are the major disadvantages and limitations of the study? Please highlight them.

Author Response

Response to Reviewer 1 Comments

Comments and Suggestions for Authors

Point 1: The authors are advised to expand the Introduction a bit by including more recent references and research findings on the topic, especially on the use of color analysis and shear force evaluation techniques for such food quality control applications.

Response 1: Thank you for your comment! Based on your comment, we have expanded the introduction part and replaced all the old references with latest ones. We have incorporated the information about color analysis and shear force evaluation techniques. The changes are made at lines 50-53, 59-70, 72, 76, 77, 81 and 84.

Point 2: It is suggested to divide the Table 1 into two tables with separate results for pH values and color parameters.

Response 2: Thank you for your comment! We have divided the Table 1 into two Tables and separate the pH values from color parameters.

Point 3: It would be better to present the results (for eg. color parameters) by bar graphs or some other plots for better visualization.

 Response 3: Thank you for your comment! We have tried to present the results in form of bar graphs but those graphs are complicated and would be tough for readers to understand, therefore, we decided to keep the data in form of Tables.

Point 4: Following papers are advised to cite in order to enhance the quality of the manuscript:

- doi: 10.1007/s13197-012-0882-x

- DOI: 10.1109/ISOEN.2017.7968861

- doi: 10.3390/ani11072111

Response 4: Thank you for your comment! We have cited two of the above-mentioned references in the manuscripts at lines 62-69 and 56 of introduction and lines 461-462 and 440 of references.

Point 5: What are the major disadvantages and limitations of the study? Please highlight them.

Response 5: Thank you for your comment! The major limitation of the current study is lack of focus on the molecular mechanisms, therefore, further studies should explore the mechanistic effect of using calcium chlo-ride to delineate the pathways involved behind improvement in meat quality attributes. The information has been included in manuscript at lines 410-413.

Reviewer 2 Report

Review comments

L1. Change the title of manuscript. What do you refer with spent buffalo bulls?

I suggest: “Evaluation of postmortem marinated with calcium chloride in meat quality of muscles M. longissimus thoracis et lumborum of young and adult buffalo bulls: ”.

L 20. Abstract: Write the summary in such a way that it indicates the statistically significant differences, and not only mentions whether the evaluated variable increased or decreased. Describe the experimental design used in this study. And how the results are displayed.

L 32. Storage properties- Maybe you refer to shelf life?

L 69-70. Delete the words in red in all cases.

L 81-82. This information is included in analyses of statistics or in the abstract. It is not necessary to indicate between parentheses. Leave only the values.

L 99-100. The pH record was obtained by three readings taken in three different areas of the sample and the average was obtained?. Specify in the methodology.

L. 133. 5 grams.

In general verify that abbreviations are mentioned before including them, p.e. CL, OA...

Author Response

Response to Reviewer 2 Comments

Comments and Suggestions for Authors

Point 1: L1. Change the title of manuscript. What do you refer with spent buffalo bulls? I suggest: “Evaluation of postmortem marinated with calcium chloride in meat quality of muscles M. longissimus thoracis et lumborum of young and adult buffalo bulls: ”.

Response 1: Thank you for your comment! Spent animals mean the animals which are no-productive dairy animals or unable to produce and reproduce and they can only use for meat purpose, therefore, in true sense these are not adult animals. The term “adult” can’t be used for these animals, as adult animals are mostly productive. The information has been included at lines 95-98. We have changed the title as “Evaluation of postmortem calcium chloride marination on meat quality characteristics of M. longissimus thoracis et lumborum of young and spent buffalo bulls”.

Point 2: L 20. Abstract: Write the summary in such a way that it indicates the statistically significant differences, and not only mentions whether the evaluated variable increased or decreased. Describe the experimental design used in this study. And how the results are displayed.

Response 2: Thank you for your comment! We have revised the abstract portion and describe the level of significance after removing the values and elaborated the experimental design and the storage days at which different parameters were measured. The changes are made at lines 22-36.

Point 3: L 32. Storage properties- Maybe you refer to shelf life?

Response 3: Thank you for your comment! We have replaced the storage properties with shelf life at line 38.

Point 4: L 69-70. Delete the words in red in all cases.

Response 4: Thank you for your comment! We have deleted all the words in red mentioned in pdf file at lines 88, 89, 102, 103, 117 and 202.

Point 5: L 81-82. This information is included in analyses of statistics or in the abstract. It is not necessary to indicate between parentheses. Leave only the values.

Response 5: Thank you for your comment! We have removed the above-mentioned information at lines 102 and 103.

Point 6: L 99-100. The pH record was obtained by three readings taken in three different areas of the sample and the average was obtained? Specify in the methodology.

Response 6: Thank you for your comment! The pH was recorded at three different points along the steaks and the average was obtained and considered as final value. It has been clarified at line 120.

Point 7: L. 133. 5 grams.

Response 7: Thank you for your comment! We have corrected the word (grams) at line 159.

Point 8: In general, verify that abbreviations are mentioned before including them, p.e. CL, OA...

Response 8: Thank you for your comment! All the abbreviations mentioned within manuscript are described before using them within text. We have rechecked it.

Reviewer 3 Report

Dear Authors,

Manuscript ID: foods-1817065
Type of manuscript: Article
Title: Effect of postmortem calcium chloride marination on meat quality
characteristics of longissimus muscles of young and spent buffalo bulls

This work deals with the evaluation of the effect of calcium chloride on the quality characteristics of buffalo bulls during storage. It is an acceptable work in this field and the obtained results could be used to improve the buffalo meat at industrial level.

In my opinion, the paper could be considered for a potential publication after major revisions. In the following, I address some suggestions which might be helpful while revising this manuscript.

Include recent references, maybe remove those references before year 2000 unless they are fundamental for the study and used in the discussion section.

Line 22-23. Please re-write the sentence including in a clear justification.

Line 41-43. Are buffalo cattle raised for meat production? Include a sentence on this topic.

Line 145. Please change the word “different” by “difference”

Line 211-247. Regarding the color of meat. Please add the Delta E characteristic as it gives important information on the effect of treatments on color of meat.

Please explain deeper the effect of calcium chloride on the meat color. Make emphasis on C*, h* and Delta E.

The papers of Goll, Taylor and Koohmaraie have been criticized due methodological errors in the shear force measurement (See this letter to the editor: https://doi.org/10.1016/S0309-1740(97)00110-1). Therefore, they are not trusty and I would not use them in this paper.

I recommend to check the following articles that would be useful for your paper.

https://doi.org/10.1108/NFS-06-2014-0058

doi: 10.1016/j.meatsci.2017.08.008

https://doi.org/10.1081/JFP-120016625

https://doi.org/10.1080/09712119.2021.1991931

https://doi.org/10.1111/j.1365-2621.2003.tb05780.x

https://doi.org/10.1016/S0309-1740(02)00201-2

https://doi.org/10.1016/S0309-1740(03)00016-0

https://doi.org/10.1016/S0309-1740(97)00083-1

International Food Research Journal 22(4): 1410-1416 (2015)

DOI: 10.1016/j.meatsci.2004.02.004

https://doi.org/10.1051/animres:2001128

https://doi.org/10.1007/s11483-018-9545-4

https://doi.org/10.1016/j.meatsci.2013.11.002

https://doi.org/10.1080/10942912.2018.1559186

DOI: 10.2527/2001.793666x

https://doi.org/10.1080/10942912.2017.1414840

https://doi.org/10.5851/kosfa.2018.38.3.515

DOI: https://doi.org/10.1017/S1751731111002047

In all Tables both, capital and lower literals should be superscripts.

Table 1. Footnote is not clear. I think it needs to say that comparisons are within each property.

Table 1. Are young and spent animals compared in table 1? I do not see the effect.

Table 4. Please move literals after the standard error.

Table 4 statistical comparison is not correct. Why some mean values have no literal? Some explanation is missing. Please check it.

It is important to add a deep discussion about Table 4. How each treatment affected the buffalo meat. Using buffalo papers would be great.

Author Response

Response to Reviewer 3 Comments

Comments and Suggestions for Authors

Dear Authors,

Manuscript ID: foods-1817065
Type of manuscript: Article
Title: Effect of postmortem calcium chloride marination on meat quality characteristics of longissimus muscles of young and spent buffalo bulls

Point 1: This work deals with the evaluation of the effect of calcium chloride on the quality characteristics of buffalo bulls during storage. It is an acceptable work in this field and the obtained results could be used to improve the buffalo meat at industrial level.

Response 1: Thank you for appreciating our efforts and for your comments and suggestions to improve the current manuscript!

Point 2: In my opinion, the paper could be considered for a potential publication after major revisions. In the following, I address some suggestions which might be helpful while revising this manuscript.

Response 2: Thank you for your suggestions! We have made changes accordingly and your suggestions really help a lot to improve the quality of the manuscript.

Point 3: Include recent references, maybe remove those references before year 2000 unless they are fundamental for the study and used in the discussion section.

Response 3: Thank you for your comment! We have replaced all the old references with latest ones especially from the introduction part of the manuscript and now we kept only those (old) references which were classical and fundamental to discuss the results of the current study. The deletion or addition of references is marked with track changes and can be seen under references section of the manuscript.

Point 4: Line 22-23. Please re-write the sentence including in a clear justification.

Response 4: Thank you for your comment! We have re-written the mentioned sentence at lines 22-23.

Point 5: Line 41-43. Are buffalo cattle raised for meat production? Include a sentence on this topic.

Response 5: Thank you for your comment! There is no any specific buffalo breed for meat production and buffalo is mainly raised for dairy purpose and when it became no-productive then sent for slaughtering. However, there are different strategies to improve meat quality characteristics in postmortem muscles. The information has been added at lines 50-53.

Point 6: Line 145. Please change the word “different” by “difference”

Response 6: Thank you for your comment! We have replaced the word “different” by “difference” at line 171.

Point 7: Line 211-247. Regarding the color of meat. Please add the Delta E characteristic as it gives important information on the effect of treatments on color of meat.

Response 7: Thank you for your comment! We have added the data of ΔE (total color change) at end of Table 2, mentioned its method of calculation at lines 131-136, described the results at lines 259-262 and discuss these results at lines 288-293.

Point 8: Please explain deeper the effect of calcium chloride on the meat color. Make emphasis on C*, h* and Delta E.

Response 8: Thank you for your comment! This time we have emphasized and discussed more on C*, h* and Δ E at lines and incorporated some new relevant references. The information has been added at lines 278-282 and 288-293 of the results and discussion part of the manuscript.

Point 9: The papers of Goll, Taylor and Koohmaraie have been criticized due methodological errors in the shear force measurement (See this letter to the editor: https://doi.org/10.1016/S0309-1740(97)00110-1). Therefore, they are not trusty and I would not use them in this paper.

Response 9: Thank you for your comment and correction! We have removed the mentioned references based on the methodological errors in it at lines 463-464.

Point 10: I recommend to check the following articles that would be useful for your paper.

https://doi.org/10.1108/NFS-06-2014-0058

doi: 10.1016/j.meatsci.2017.08.008

https://doi.org/10.1081/JFP-120016625

https://doi.org/10.1080/09712119.2021.1991931

https://doi.org/10.1111/j.1365-2621.2003.tb05780.x

https://doi.org/10.1016/S0309-1740(02)00201-2

https://doi.org/10.1016/S0309-1740(03)00016-0

https://doi.org/10.1016/S0309-1740(97)00083-1

International Food Research Journal 22(4): 1410-1416 (2015)

DOI: 10.1016/j.meatsci.2004.02.004

https://doi.org/10.1051/animres:2001128

https://doi.org/10.1007/s11483-018-9545-4

https://doi.org/10.1016/j.meatsci.2013.11.002

https://doi.org/10.1080/10942912.2018.1559186

DOI: 10.2527/2001.793666x

https://doi.org/10.1080/10942912.2017.1414840

https://doi.org/10.5851/kosfa.2018.38.3.515

DOI: https://doi.org/10.1017/S1751731111002047

Response 10: Thank you for recommending the relevant references! We have included these references (doi: 10.1016/j.meatsci.2017.08.008; DOI: 10.1081/JFP-200059476; https://doi.org/10.1111/j.1365-2621.2003.tb05780.x; https://doi.org/10.1016/S0309-1740(03)00016-0; DOI: 10.2527/2001.793666x; DOI: https://doi.org/10.1017/S1751731111002047) and some of these references were already present within text (https://doi.org/10.1108/NFS-06-2014-0058; DOI: 10.1016/j.meatsci.2004.02.004). In total we have removed 10 old references and included 15 latest references within the manuscript.

Point 11: In all Tables both, capital and lower literals should be superscripts.

Response 11: Thank you for your comment! We have moved the subscripts to superscripts and now all the letters are superscript in all Tables.

Point 12: Table 1. Footnote is not clear. I think it needs to say that comparisons are within each property.

Response 12: Thank you for your comment! We have clarified the footnotes of all the Tables based on your comment at lines 223-224, 265-267, 325-327, 361-363 and 397-398.

Point 13: Table 1. Are young and spent animals compared in table 1? I do not see the effect.

Response 13: Thank you for your comment! Yes the young and spent buffalo bulls were compared in Table 1 (now Tables 1 and 2), it can be seen in the same column with different capital letters (A,B,C…) within storage days.

Point 14: Table 4. Please move literals after the standard error.

Response 14: Thank you for your comment! We have moved all the literals after standard error in Table 4 (now Table 5).

Point 15: Table 4 statistical comparison is not correct. Why some mean values have no literal? Some explanation is missing. Please check it.

Response 15: Thank you for your comment! We have rechecked and the statistical comparison is correct. The values which were not significantly different were not assigned any letters, however, in order to avoid confusion, we have allotted them the same letters, indication insignificant difference between them.

Point 16: It is important to add a deep discussion about Table 4. How each treatment affected the buffalo meat. Using buffalo papers would be great.

Response 16: Thank you for your comment! All the parameters mentioned in the Table 4 (now Table 5) are discussed in depth at their corresponding section of the discussion. However, some of the parameters which were weakly described (such as C* and h values) previously are now discussed in details.

Round 2

Reviewer 3 Report

Dear Authors,

Thank you to attend all suggested modifications in your manuscript.

It has improved significantly and no further changes are needed. 

Congratulations.

Author Response

Thank you very much for your appreciations and accepting our article.